



# The impact of large-scale circulation on daily fine particulate matter (PM₂.₅) over major populated regions of China in winter

Zixuan Jia[1], Ruth M. Doherty[1], Carlos Ordóñez[2], Chaofan Li[3,6], Oliver Wild[4], Shipra Jain[1], Xiao Tang[5]

[1]School of GeoSciences, University of Edinburgh, Edinburgh, UK
[2]Departamento de Física de la Tierra y Astrofísica, Facultad de Ciencias Físicas, Universidad Complutense de Madrid, Madrid, Spain
[3]Center for Monsoon System Research, Institute of Atmospheric Physics, Chinese Academy of Sciences, Beijing, China
[4]Lancaster Environment Centre, Lancaster University, Lancaster, UK
[5]LAPC, Institute of Atmospheric Physics, Chinese Academy of Sciences, Beijing, China
[6]College of Earth and Planetary Sciences, University of the Chinese Academy of Sciences, Beijing, China

*Correspondence to*: Zixuan Jia (Z.Jia-6@sms.ed.ac.uk)

**Abstract.** The influence of large-scale circulation on daily PM₂.₅ variability through its direct effect on key regional meteorological variables is examined over three major populated regions of China: Beijing–Tianjin–Hebei (BTH), the Yangtze River Delta (YRD), and the Pearl River Delta (PRD). In BTH, a shallow East Asian trough curbs northerly cold and dry air from the Siberian High, enhancing PM₂.₅ pollution levels. Weak southerly winds in eastern and southern China, associated with a weakened Siberian High, suppress horizontal dispersion, contributing to air pollution accumulation over YRD. In PRD, weak southerly winds and precipitation deficits over southern China are conducive to high PM₂.₅ pollution levels. To account for these dominant large-scale circulation – PM₂.₅ relationships, we propose three new circulation-based indices for predicting different levels of air pollution based on regional PM₂.₅ concentrations in each region: a 500 hPa geopotential height-based index for BTH, a sea level pressure-based index for YRD and an 850 hPa meridional wind-based index for PRD. These three indices can effectively distinguish clean days from heavily polluted days in these regions, assuming variation is solely due to meteorology. We also find that including the most important regional meteorological variable in each region improves the performance of the circulation-based indices in predicting daily PM₂.₅ concentrations on the regional scale. These results are beneficial to understanding and forecasting the occurrence of heavily polluted PM₂.₅ days in BTH, YRD and PRD from a large-scale perspective.

## 1 Introduction

Over the last few decades, rapid economic progress and urbanization in China have raised a number of environmental challenges. These include sharp increases in the atmospheric concentrations of particulate matter with an aerodynamic diameter of 2.5 μm or less (PM₂.₅), which are of utmost concern for public health (Xu et al., 2013; Zheng et al., 2015). Episodes of haze and smog pollution with high levels of PM₂.₅, in particular during winter, have become common in the most developed and highly populated city clusters in China (Zhang et al., 2007; Chan and Yao, 2008; Zhang et al., 2014).



Although emissions of pollutant precursors strongly influence air pollution levels, meteorology also plays a major role in air quality variability and trends through a combination of transport, transformation and deposition processes (e.g., Wang et al., 2009; Cheng et al., 2019). For instance, the extreme haze event in January 2013 in Beijing when the maximum instantaneous

$PM_{2.5}$ value exceeded 500 µg/m³, one of the worst air pollution events on record in China, has been attributed to unfavourable atmospheric dispersion conditions with weak surface winds and high humidity (Wang et al., 2014; Yang et al., 2015). In contrast, during winter and spring in 2015, $PM_{2.5}$ concentrations were much lower at most Chinese monitoring stations because of more favourable atmospheric dispersion conditions compared to those of the previous year (Wang et al., 2016).

While regional meteorological conditions are known to strongly influence air pollution levels, the responses of $PM_{2.5}$ concentrations to different meteorological variables are complex (e.g., Tai et al., 2010; Barmpadimos et al., 2012; Dawson et al., 2014; Han et al. 2016; Garrido-Perez et al., 2017, 2021). Key processes and the relevant regional meteorological variables influencing $PM_{2.5}$ levels have been identified in recent studies. These processes include: a) secondary aerosol formation and hygroscopic growth associated with high relative humidity (RH; Sun et al., 2013; Wang et al., 2014), b) wet

deposition due to precipitation (Koch et al., 2003; Tai et al., 2010), c) horizontal dispersion of polluted air under high wind speeds (WSPD; Wang et al., 2012; Zhang et al., 2014), and d) vertical ventilation and dilution of the boundary layer via mechanically generated turbulence associated with strong wind shear (WSHR; Wang et al., 2018, 2019a) and via thermodynamically generated turbulence as measured by inversion intensity (INV; Zhao et al., 2013; Wang et al., 2014). Specifically, high RH, weak WSPD, strong INV and weak WSHR have been found to contribute to the accumulation and

growth of pollutants in a shallow and stable boundary layer over the North China Plain (e.g., Wang et al., 2014; Ge et al., 2019). However, RH can also be associated with precipitation and therefore removal of aerosols by wet deposition (Zhu et al., 2012; Leung et al., 2018), and may also be an indicator of air masses from different origins.

These key regional meteorological factors have been found to be affected by circulation patterns at larger scales over different regions of the globe (Tai et al., 2012; Garrido-Perez et al., 2017; Pei et al., 2018). Prominent large-scale circulation

patterns over China during winter include the East Asian winter monsoon (EAWM; Chang et al., 2006; Wang and Chen, 2010) and El Niño-Southern Oscillation (ENSO; Wang et al., 2000; Zhang et al., 2017). The EAWM resulting from the development of the cold-core Siberian High system is mainly characterised by dry cold low-level northerlies along the East Asian coast, the mid-tropospheric East Asian trough and the upper-tropospheric westerly jet stream (Jhun and Lee, 2004; Li and Yang, 2010; Wang and Lu, 2017). The EAWM has a significant impact on China's regional meteorological conditions,

including air temperature, wind speed, RH and atmospheric stability (Jeong and Park, 2017; Wang et al., 2019b), and hence influences $PM_{2.5}$ levels as noted above. ENSO, as the dominant mode of global ocean-atmosphere interaction, also substantially modulates regional meteorological conditions in China, through changes in atmospheric circulation patterns. The regional meteorological variables affected include RH and precipitation over southeastern China, and wind speed over northern China (Sun et al., 2018; He et al., 2019).



Previous studies of how the large-scale wintertime circulation modulates air quality in China through its effect on regional meteorology have been primarily focused on Beijing and the North China Plain, the regions with the most severe $PM_{2.5}$ pollution (e.g., Wang et al., 2014, 2019b; Zhang et al., 2019). Broader regions in northern and southern China also show clear relationships between $PM_{2.5}$ concentrations and the EAWM intensity (e.g., Jeong and Park, 2017), aerosol optical depth and the position of the Siberian High (Jia et al., 2015), as well as the number of haze days and ENSO intensity (e.g., He et

al., 2019). However, the major city clusters in northern, eastern and southern China, i.e., Beijing–Tianjin–Hebei (BTH), the Yangtze River Delta (YRD) and the Pearl River Delta (PRD), respectively, have been considered jointly only in a few studies (e.g., Leung et al., 2018; Hou et al., 2019). Furthermore, most of the existing large-scale circulation indices, such as the EAWM indices (Wang et al., 2010), the Siberian High index (Wu and Wang, 2002) and the Haze Weather Index (Cai et al., 2017), have been proposed for the North China Plain. Consequently, they do not reflect the link between the large-scale

circulation and $PM_{2.5}$ levels over YRD and PRD. Indeed, Leung et al. (2018) found using $PM_{2.5}$ observations between June 2014 and May 2017 that different dominant large-scale circulation modes could explain annual $PM_{2.5}$ variability in BTH, YRD and PRD.

Understanding the impact of the large-scale circulation on $PM_{2.5}$ air quality in these three major populated regions of China during winter, therefore, requires consideration of regional differences in the dominant large-scale circulation features. In

order to understand and predict the occurrence of days with high $PM_{2.5}$ concentrations, it is critical to investigate the relationship between the large-scale circulation and $PM_{2.5}$ levels on daily timescales. This study examines the dominant large-scale circulation-$PM_{2.5}$ relationships separately for BTH, YRD and PRD during winter, and further proposes novel circulation-based indices to explain the day-to-day variability of $PM_{2.5}$ levels in each region. We first explore the relationship of daily $PM_{2.5}$ concentrations with specific regional meteorological variables across BTH, YRD and PRD (Section 3). We

then identify the dominant large-scale circulation associated with heavily polluted days for the three regions through its effect on the most important regional meteorological variables, and propose specific circulation-based indices for these three regions (Section 4). Furthermore, we assess the performance of these circulation-based indices in distinguishing different levels of air pollution (Section 5) and examine the joint effect of the circulation-based indices and regional meteorology on the day-to-day variability of $PM_{2.5}$ (Section 6). Finally, Section 7 summarises the main results.

**2 Data and Methodology**

We use daily meteorological data from the ECMWF Reanalysis 5 (ERA5; Hersbach et al., 2020) on a 30 km × 30 km grid, including zonal wind at 300 hPa, 900 hPa and 1000 hPa (U300, U900, U1000); meridional wind at 850 hPa, 900 hPa and 1000 hPa (V850, V900, V1000); geopotential height at 500 hPa (Z500); air temperature at 900 hPa and 1000 hPa; RH at 900 hPa and 1000 hPa; sea level pressure (SLP) and sea surface temperatures (SSTs). Hourly data are used to calculate daily

averages for 450 days during the five winters from 1st Dec 2013 – 28th Feb 2014 to 1st Dec 2017 – 28th Feb 2018 (hereafter referred to as DJF 2013–2017). Daily precipitation is from the Global Precipitation Climatology Project (GPCP; Huffman et al. 2001) 1° daily precipitation product. These meteorological fields are used to investigate both the large-scale circulation





features and regional meteorological conditions modulating PM$_{2.5}$ concentrations. Four meteorological fields representing relevant processes affecting PM$_{2.5}$ in the boundary layer (RH, WSPD, WSHR and INV) are evaluated, following Ge et al.
(2019). RH and WSPD are used at 1000 hPa. Wind shear, WSHR, is calculated as

$$\text{WSHR} = \sqrt{(\text{U900} - \text{U1000})^2 + (\text{V900} - \text{V1000})^2},\tag{1}$$

Inversion intensity, INV, is calculated as

$$\text{INV} = \theta_{v,900\,\text{hPa}} - \theta_{v,1000\,\text{hPa}},\tag{2}$$

where $\theta_v$ is virtual potential temperature and the subscripts 900 and 1000 hPa specify the vertical levels at which $\theta_v$ is
evaluated from air temperature and RH.

The six-year long high-resolution Chinese air quality reanalysis dataset (CAQRA; Kong et al., 2021) is the latest long-term air quality reanalysis for China. It contains surface fields of conventional pollutants, including PM$_{2.5}$, at high spatial (15 km×15 km) and temporal (1 h) resolution for the period 2013–2018. This dataset has been developed by assimilating pollutant concentrations from over 1000 surface air quality monitoring sites from the China National Environmental
Monitoring Centre. CAQRA has been validated against independent datasets, yielding a good performance in reproducing the magnitude and variability of surface air pollutants in China on a regional scale (Kong et al., 2021). We use PM$_{2.5}$ hourly concentrations from this dataset to calculate daily averages for the same time period as the daily meteorological data (DJF 2013–2017, 450 days). PM$_{2.5}$ concentrations show a decreasing trend over the period of analysis, consistent with the primary emission reductions and PM$_{2.5}$ concentration decreases reported by many previous studies (e.g., Li et al., 2019, 2020).
Therefore, to eliminate the influence of changing anthropogenic emissions, the daily PM$_{2.5}$ data are de-trended by removing the linear trend from the Dec 2013–Feb 2018 (1550 days) time series. To understand how meteorology drives clean vs. polluted conditions in a consistent way, percentile thresholds of the de-trended daily PM$_{2.5}$ data are used. We choose the 10[th] percentile (p10) of PM$_{2.5}$ concentrations as the clean threshold and the 90[th] percentile (p90) of PM$_{2.5}$ concentrations as the heavily polluted threshold. We then group all the days below p10 and above p90 and classify them as clean or heavily
polluted days (45 days each).

Statistical significance is assessed at the 95% confidence level throughout this paper, unless otherwise stated. The effective numbers of degrees of freedom are calculated in order to assess the significance of correlations considering the effect of temporal autocorrelation (Allen and Smith 1994; Hu et al. 2017). A non-parametric bootstrap resampling method is used to assess the significance of differences between meteorological variables under heavily polluted and average conditions, as
these variables do not necessarily follow normal distributions. This bootstrap resampling method generates random samples of meteorological variables for the whole period of analysis. Each random sample comprises 45 days, i.e., the total number of heavily polluted days. Then the difference between the mean of each sample and all the data is calculated. This procedure is repeated 10,000 times to create a random distribution of meteorological variable differences. Following this, differences calculated for heavily polluted days are compared with the distribution of meteorological variable differences. The



differences calculated for heavily polluted days are considered significantly negative or positive (at 95% confidence level) when they are below or above the 2.5% and 97.5% tails, respectively.

## 3 Influence of regional meteorological variables on daily PM$_{2.5}$ variability

We first identify the meteorologically coherent regions representing BTH, YRD and PRD by searching for reanalysis grid cells where the daily PM$_{2.5}$ concentrations are highly correlated ($r \geq 0.7$) with those in the grid cells corresponding to

Beijing, Shanghai and Guangzhou, respectively (Fig. 1). This accounts for the regional nature of PM$_{2.5}$ pollution and provides a more robust result than using the closest grid cells containing the cities or some arbitrary rectangular regions as previous studies have done (e.g., Leung et al., 2018; Hou et al., 2019). Daily regional PM$_{2.5}$ concentrations are then calculated by averaging the data over these three homogeneous regions. Note that, as the 90[th] percentiles (p90) of daily average PM$_{2.5}$ differ for the three regions, heavily polluted days defined on p90 correspond to concentrations > 97 µg/m$^3$ for

BTH, > 110 µg/m$^3$ for YRD and > 68 µg/m$^3$ for PRD. The value of p90 PM$_{2.5}$ is higher in YRD than in BTH, because the smaller size of YRD is more representative of a coherent urban environment (Fig. 1). For consistency, the gridded meteorological fields described in Section 2 are averaged over the same regions to construct daily regional meteorological variables.

Figure 2 shows the lagged relationship of daily regional PM$_{2.5}$ concentrations with specific regional meteorological variables

in these three homogeneous regions for the entire DJF 2013–2017 period. There are positive correlations for INV and negative correlations for WSHR and WSPD with PM$_{2.5}$ concentrations for all three regions. This occurs even when daily PM$_{2.5}$ concentrations are lagged by a few days. This suggests that high PM$_{2.5}$ days are associated with poor vertical ventilation (increased INV and reduced WSHR) and reduced horizontal dispersion (weak WSPD) for several days preceding the high PM$_{2.5}$ levels. In particular, WSPD is the variable with the highest correlation with PM$_{2.5}$ concentrations in YRD,

appearing for a one-day lag ($r = -0.43$) (Fig. 2b). Unlike the other three variables considered, the relationship between RH and PM$_{2.5}$ concentration varies across BTH, YRD and PRD. A positive correlation is seen between RH and PM$_{2.5}$ concentrations for BTH, with the highest value at zero lag ($r = 0.66$) (Fig. 2a). This highlights the general contrast between clean, dry air reaching BTH from the northwest and more polluted, humid air reaching BTH from central and eastern China. However, RH is negatively correlated with PM$_{2.5}$ concentrations in the other two regions, with larger correlations in PRD

than in YRD. The high correlations in PRD persist over the previous four days (with the highest value of $r = -0.52$ for a two-day lag) (Fig. 2c). This reflects the association of high RH with cleaner oceanic air and precipitation, and hence wet deposition in PRD (e.g., Zhu et al., 2012; Jeong and Park, 2017). RH is the meteorological variable presenting the highest correlation value with PM$_{2.5}$ concentrations over both BTH and PRD. These results are consistent with previous findings of the different patterns in PM$_{2.5}$–RH relationships over northern and southern China (Wang et al., 2014; Leung et al., 2018).

Consequently, RH on the same day ($r = 0.66$), WSPD one day before ($r = -0.43$) and RH two days before ($r = -0.52$) are identified as the most important regional meteorological variables contributing to the day-to-day variability of PM$_{2.5}$ concentrations over BTH, YRD and PRD, respectively. Among the second most relevant meteorological variables, WSPD





and INV stand out for BTH and PRD, respectively, with absolute correlation coefficients close to 0.5 for some time lags. Following previous analyses (e.g., Tai et al., 2010, 2012; Leung et al., 2018; Ge et al., 2019), we now investigate how the
relationships between PM$_{2.5}$ concentrations and the most important regional meteorological variables described above and considering the same time lags are caused by common association with large-scale circulation systems.

**4 Modulation of daily PM$_{2.5}$ by the large-scale circulation**

Using ERA-5 reanalysis data for DJF 2013-17, we find the wintertime large-scale circulation over East Asia is dominated by the Siberian High as seen from the high SLP values centred over northwestern Mongolia (Fig. 3a). The Siberian High
induces northerly near-surface winds along its eastern edge, which bring cold, clean air to northern and central China as indicated by negative V850 values (Fig. 3b). This northerly near-surface flow is also associated with the middle tropospheric East Asian trough, characterised by low Z500 values over Northeast China as seen in Figure 3c. Over eastern and southern China, wet and warm southerly winds blow from the South China Sea (Fig. 3b), bringing precipitation (Fig. 3d).

Previous studies have introduced a variety of large-scale circulation indices to characterise atmospheric circulation in East
Asia. Here we apply three commonly used EAWM indices ($I_{Yang}$ (V850): Yang et al., 2002; $I_{Sun}$ (Z500): Sun and Li, 1997; $I_{Jhun}$ (U300): Jhun and Lee, 2004) and a widely used Siberian High index ($I_{SH}$; Wu and Wang, 2002) to test their relationship with daily PM$_{2.5}$ concentrations separately for the three meteorologically coherent regions using reanalysis data (Table S1). We reverse $I_{Yang}$ and $I_{Sun}$ by multiplying them by $-1$ so that a high index value represents a strong EAWM. The three EAWM indices have been selected because they reflect the circulation characteristics of the EAWM in the lower, middle and upper
troposphere, respectively (e.g., Wang et al., 2019b). Linear correlations of all three EAWM indices with the daily PM$_{2.5}$ concentrations for the whole period of analysis are significant (at 99% confidence level) only for BTH ($r$ ranging from -0.54 to -0.36), whereas absolute correlation coefficients do not exceed 0.12 for YRD and PRD. This suggests that these three typical EAWM indices do not capture well the relationship between the large-scale circulation and daily PM$_{2.5}$ concentrations over the YRD and PRD regions used in this study. The Siberian High index ($I_{SH}$) is significantly correlated
with daily PM$_{2.5}$ concentrations for all three regions, although the correlations are not strong ($r$ ranging from -0.19 to -0.13).

Considering the moderate correlations found for YRD and PRD, we further investigate the influence of large-scale circulation on daily PM$_{2.5}$ variability through its direct effect on the most important regional meteorological variables identified separately for the three regions. For this purpose, we first examine the dominant large-scale circulation features associated with heavily polluted days for each region, then identify the correlation patterns of daily PM$_{2.5}$ concentrations
with these circulation variables for the whole period of analysis and define circulation-based indices separately for the three regions. These analyses will be carried out considering the same time lags as those for the most important regional meteorological variables identified in Section 3. The daily meteorological reanalysis data are normalised by subtracting the means of individual variables and dividing by their standard deviations to yield fields with zero means and unit variance before calculating these indices.



## 4.1 Beijing–Tianjin–Hebei (BTH)

As shown in Figure 2a, the strongest correlations between daily PM$_{2.5}$ concentrations and regional meteorological variables over BTH are found for RH with no time lag. In this section, we examine circulation variables during heavily polluted days (PM$_{2.5}$ above p90; daily PM$_{2.5}$ concentrations > 97 μg/m$^3$ for BTH) over this region. Figure 4 shows the average composites of circulation variables (SLP, V850 and Z500) for heavily polluted days over BTH (upper panels), along with the difference (lower panels) between heavily polluted days and the winter (DJF) mean (as displayed in Figure 3) during 2013–2017. Heavily polluted days are characterised by a weak and eastward-extended Siberian High, weak northerly winds at 850 hPa over North China, and a shallow East Asian trough at 500 hPa, reflecting a weak EAWM circulation (Jia et al., 2015; Ge et al., 2019). Following these results, we calculate daily correlations of the PM$_{2.5}$ concentrations with SLP, V850 and Z500 for the whole period of analysis to assess to what extent the observed circulation anomalies can be used to represent the day-to-day variability of PM$_{2.5}$. The resulting circulation-PM$_{2.5}$ correlation patterns are displayed in Fig. 5. The daily PM$_{2.5}$ concentrations for BTH have negative correlations with SLP over mainland China (and positive correlations centred over Japan, Fig. 5a), positive correlations with V850 over eastern China (Fig. 5b), and positive correlations with Z500 centred over Northeast China (Fig. 5c), in accord with the observed departures of heavily polluted days from the winter mean.

Based on these circulation-PM$_{2.5}$ correlation patterns, we now select broad regions (yellow rectangles in Figure 5) which represent the highest correlations with PM$_{2.5}$ concentrations in BTH and then construct spatial averages of the daily meteorological fields over these regions. The area-weighted averages of daily normalised SLP, V850 and Z500 show significant correlations with daily PM$_{2.5}$ concentrations in BTH (at 99% confidence level), especially for Z500 ($r = 0.67$), followed by V850 ($r = 0.59$) and SLP ($r = 0.54$) (Table 1). Note that these correlations are stronger than those using the EAWM indices and the Siberian High index from the literature (see Table S1). We therefore use Z500 averaged over Northeast China, Korea and the Sea of Japan [118–139°E, 33–50°N] (rectangle in Figure 5c) to build a Z500-based index for BTH ($I_{Z500\_BTH}$) for all days in DJF 2013-17). $I_{Z500\_BTH}$ is calculated as the mean of daily normalised Z500 in that region with a reversed sign (eq. 3) so that negative values of $I_{Z500\_BTH}$ indicate a shallow East Asian trough.

$$I_{Z500\_BTH} = - \overline{Z500\ (33°-50°N, 118°-139°E)} \tag{3}$$

$I_{Z500\_BTH}$ is significantly correlated both with PM$_{2.5}$ concentrations ($r = -0.67$ in Table 2) and with RH ($r = -0.64$ in Table 2) in BTH on daily time scales. These results point to a shallow East Asian trough as the dominant large-scale circulation pattern favouring high PM$_{2.5}$ concentrations and high RH in BTH. The shallow East Asian trough in the middle troposphere inhibits the invasion of northerly cold air from the rear of the trough to northern and central China, yielding southerly wind anomalies (Figs. 4e-f), as found in other studies (e.g., Zhang et al., 2014). This anomalous warm and humid air from the south therefore creates appropriate conditions for the accumulation and possibly the growth of fine aerosols and also suppresses the southward transport of aerosols away from BTH (see positive correlations for RH and negative correlations for WSPD in Fig. 2a).





## 4.2 Yangtze River Delta (YRD)

As shown in Figure 2b, the correlations between daily PM$_{2.5}$ concentrations and regional meteorological variables over YRD
are highest for the most important regional meteorological variable (WSPD) when daily PM$_{2.5}$ concentrations are lagged by
one day. Hence, in this section, we focus on the circulation variables (SLP, V850 and Z500) one day before heavily polluted
days over this region. Heavily polluted days in YRD (PM$_{2.5}$ above p90; daily PM$_{2.5}$ concentrations > 110 μg/m$^3$) are mainly
characterised by reduced SLP over eastern China, indicating a weak Siberian High (Figs. 6a and 6d), and a shallow East
Asian trough with positive Z500 anomalies centred over Japan (Figs. 6c and 6f). This weakened intensity of the Siberian
High is associated with a northerly wind anomaly over both North and South China, as well as a significant southerly wind
anomaly over Northeast China and Japan (Fig. 6e). The northerly wind anomaly implies a weakening of the winter mean
southerly wind over southern China and a strengthening of the winter mean northerly wind over northern China (Figs. 6b,
6e). This different pattern in southern versus northern China is further supported by the daily WSPD-PM$_{2.5}$ correlation
features for the whole period of analysis, where daily PM$_{2.5}$ concentrations in YRD are negatively correlated with WSPD
over southern China and positively correlated over northern China (Fig.S1). Furthermore, the daily PM$_{2.5}$ concentrations for
YRD have negative correlations with SLP centred over northeast China (Fig. 7a), negative correlations with V850 over both
southern China and northern China (and positive correlations over northeast China and Japan, Fig. 7b), and positive
correlations with Z500 centred over northwest China (Fig. 7c). These circulation-PM$_{2.5}$ correlation patterns for the whole
period of analysis are consistent with the circulation anomalies shown for heavily polluted days in Figure 6.
We then identify the regions with the highest correlations of area-weighted average daily normalised meteorological fields
with daily PM$_{2.5}$ concentrations in YRD. Among these three meteorological fields (i.e., SLP, V850, Z500), for the regions
that show the highest correlations with PM$_{2.5}$ concentrations in YRD (yellow rectangles in Fig 7), SLP is found to have the
highest correlation ($r$ = -0.33) (Table 1). We therefore use SLP averaged over Northeast China [30–49°N, 111–131°E]
(rectangle in Figure 7a) to build a normalised SLP-based index for YRD ($I_{SLP\_YRD}$) for all days in DJF 2013-17 (eq. 4).
Negative values of $I_{SLP\_YRD}$ indicate a weak Siberian High.

$$I_{SLP\_YRD} = \overline{SLP\ (30°-49°N, 111°-131°E)} \qquad (4)$$

$I_{SLP\_YRD}$ is significantly correlated both with PM$_{2.5}$ concentrations ($r$ = -0.33 in Table 2) and with WSPD ($r$ = 0.29 in Table 2)
in YRD on daily time scales. This suggests a weakened Siberian High as the dominant large-scale circulation pattern
contributing to higher concentrations of PM$_{2.5}$ and reduced WSPD in YRD. The associated reduction in the southerly wind
reported above for southern and eastern China together with reduced WSPD implies a greater suppression of horizontal
dispersion, contributing to air pollution accumulation over YRD. Moreover, strengthened northerly winds in northern China
may lead to southward transport of aerosols emitted from sources over northern China to YRD, as also indicated by previous
studies (Li et al., 2012; Jeong and Park, 2017).



We repeated the analysis above to examine the sensitivity to different time lags. The observed circulation anomaly patterns without a lag resemble those found for one-day lag, although they are displaced to the east because of the eastward movement of synoptic systems in the midlatitudes (Fig. S2). The region that shows the highest correlations with PM$_{2.5}$ concentrations in YRD on the SLP-PM$_{2.5}$ correlation pattern is slightly less significant without a lag, again with an eastward shift (Fig. S3).

**4.3 Pearl River Delta (PRD)**

In contrast to BTH and YRD, the highest correlations of daily PM$_{2.5}$ concentrations over PRD with the two most important regional meteorological variables (RH and INV) persist when PM$_{2.5}$ is lagged by several days (Fig. 2c). As the maximum correlations are found with a lag of two days, we examine composites of two circulation variables (SLP and V850) and precipitation two days before the occurrence of heavily polluted days over PRD (PM$_{2.5}$ above p90; daily PM$_{2.5}$

concentrations > 68 µg/m$^3$) (Fig. 8). These are mainly characterised by reduced SLP centred over northern China and increased SLP over southwestern China, weak southerly winds at 850 hPa over South China, as well as precipitation deficits over southern China. Correlation patterns of PM$_{2.5}$ with the same fields (Fig. 9) for the whole period of analysis further support these circulation anomalies for heavily polluted days. Daily PM$_{2.5}$ concentrations over PRD have negative correlations with SLP over northern China (and positive correlations over southern China, Fig. 9a), negative correlations

with V850 over South China and the South China Sea (Fig. 9b), as well as with precipitation over southern China (Fig. 9c). There are also negative correlations between daily PM$_{2.5}$ concentrations and SSTs over the central and eastern equatorial Pacific (and positive correlations over the western equatorial Pacific), as well as negative correlations for SLP over the western North Pacific (Fig. S4). These circulation-PM$_{2.5}$ correlation features display characteristic ENSO-related patterns over the Pacific and East Asia (e.g., Wang et al., 2000). La Niña events are associated with warm SSTs in the western Pacific

and cold SSTs in the central and eastern equatorial Pacific, reduced SLP over the western North Pacific and descending motion on the northwestern flank of this reduced SLP. The opposite relationships are seen for El Niño (Fig. S4). This anomalous subsidence with suppressed precipitation (Fig. 9) has been found to play a major role in high PM$_{2.5}$ concentrations over southern China (e.g., He et al., 2019; Sun et al., 2018). We also find that more than 80% (37 out of 45) of heavily polluted days in PRD are in La Niña years, considered here as those when the Niño 3.4 index (area-weighted averages of

SSTs anomaly over 5°S –5°N, 120° –170°W) is less than -0.5. Nonetheless, these results should be treated with caution because of the relatively short time series considered (only 5 winters with PM$_{2.5}$ data).

Comparing the correlations of the area-weighted average daily normalised meteorological fields with daily PM$_{2.5}$ concentrations, V850 is found to have the highest value ($r$ = -0.43 in Table 1), followed by SLP and precipitation ($r$ <0.4) over the regions showing the highest correlation with PM$_{2.5}$ concentrations in PRD (yellow rectangles in Figure 9). We

therefore build a normalised daily V850-based index for PRD ($I_{V850\_PRD}$) by averaging V850 over the region of South China and the South China Sea [100–118°E, 10–22°N] (rectangle in Fig. 9b) (eq. 5). Negative values of $I_{V850\_PRD}$ indicate weak southerly winds over South China.



$$I_{V850\_PRD} = \overline{V850\ (10°\!-\!22°N, 100°\!-\!118°E)} \tag{5}$$

Weak southerly winds over southern China as the dominant large-scale circulation pattern are associated with greater polluted continental flow and precipitation deficits under weak cleaner oceanic winds (Figs. 8e-f) that are conducive to air pollution over PRD via reduced wet deposition. Consequently, $I_{V850\_PRD}$ is not only negatively correlated with PM$_{2.5}$ concentrations ($r$ = -0.43 in Table 2) but also positively correlated with regional RH in PRD ($r$ = 0.64 in Table 2). The anomalous subsidence yielding precipitation deficits over southern China is also associated with a shallow and stable

boundary layer where the vertical dilution capacity of the lower atmosphere reduces (see negative correlations for RH and positive correlations for INV in Fig. 2c). Overall, the observed circulation patterns for smaller and zero lag are broadly similar to those found for a two-day lag (Fig. S5), although the V850-PM$_{2.5}$ correlations weaken as the lag is reduced (Fig. S6).

## 5 Performance of circulation-based indices for differing air pollution levels

Our analyses confirm that the proposed circulation-based indices are significantly correlated with the most important regional meteorological variables and the PM$_{2.5}$ concentrations on daily time scales during DJF 2013-17. The correlations are significant at the 99% confidence level (Table 2). To further examine the performance of circulation-based indices for distinguishing different levels of air quality, we show the distributions of $I_{Z500\_BTH}$, $I_{SLP\_YRD}$ and $I_{V850\_PRD}$ for several percentile thresholds of daily PM$_{2.5}$: above p90 (heavily polluted), p50-90 (moderately polluted), p10-50 (moderately clean) and below

p10 (clean) (Fig. 10). Note that the sample size for moderate events is larger than for heavily polluted/clean events and also that daily PM$_{2.5}$ concentrations are lagged by one and two days in the case of YRD and PRD, respectively, for consistency with the previous analysis.

For BTH, the average value of $I_{Z500\_BTH}$ with associated 95% confidence intervals are: $I_{Z500\_BTH}$ = -1.04 $\pm$ 0.20 for heavily polluted days, $I_{Z500\_BTH}$ = -0.28 $\pm$ 0.10 for moderately polluted days, $I_{Z500\_BTH}$ = 0.35 $\pm$ 0.10 for moderately clean days and

$I_{Z500\_BTH}$ = 0.83 $\pm$ 0.23 for clean days (Fig. 10a). The values of $I_{Z500\_BTH}$ for these four categories differ (i.e., the confidence intervals do not overlap) at the 95% confidence level and $I_{Z500\_BTH}$ can distinguish between different levels of air quality, not just extreme heavily polluted or clean conditions. Ge et al. (2019) used a Siberian High index *($I_{SH}$*; Wu and Wang, 2002), which we tested in Section 4, and a potential vorticity based EAWM index *($I_{PV}$*; Huang et al., 2016) to distinguish different PM$_{2.5}$ pollution levels in Beijing. They found that $I_{SH}$ can effectively distinguish clean days (daily PM$_{2.5}$ concentrations $\leq$ 75

μg/m$^3$) from polluted days (daily PM$_{2.5}$ concentrations $\geq$ 75 μg/m$^3$), but could not distinguish between moderate and severe (daily concentrations PM$_{2.5}$ $\geq$ 150 μg/m$^3$) PM$_{2.5}$ pollution. The $I_{PV}$ index exhibited the reverse problem. This shows that $I_{Z500\_BTH}$ performs better than existing circulation indices, both in capturing the relationship between the dominant large-scale circulation and daily PM$_{2.5}$ concentrations (Tables S1 and 2) and in distinguishing pollution levels in BTH (Fig. 10a). In the case of YRD (Fig. 10b), $I_{SLP\_YRD}$ can effectively distinguish heavily polluted days ($I_{SLP\_YRD}$ = -0.32 $\pm$ 0.28) from clean days

($I_{SLP\_YRD}$ = 0.51 $\pm$ 0.19). However, differences are not significant between heavily and moderately polluted days ($I_{SLP\_YRD}$ = -



0.19 $\pm$ 0.11) and are not highly significant between clean and moderately clean days ($I_{SLP\_YRD}$ = 0.24 $\pm$ 0.11). For PRD (Fig. 10c), $I_{V850\_PRD}$ can distinguish well between heavily polluted days ($I_{V850\_PRD}$ = -0.31 $\pm$ 0.16), moderately clean days ($I_{V850\_PRD}$ = 0.22 $\pm$ 0.10) and clean days ($I_{V850\_PRD}$ = 0.83 $\pm$ 0.19), but not between heavily polluted and moderately polluted days ($I_{V850\_PRD}$ = -0.28 $\pm$ 0.09).

To further illustrate the relationships between the dominant large-scale circulation, as represented by these circulation-based indices, and the severity of $PM_{2.5}$ pollution at daily timescales, we show the joint frequency distributions of daily values of circulation-based indices compared to daily $PM_{2.5}$ concentrations (Fig. 11). We show the linear relationship between each respective index and $PM_{2.5}$ concentrations, as given in Table 2, with higher $PM_{2.5}$ concentrations and smaller (negative) index values on heavily polluted days, and vice versa. Moderately polluted days ($PM_{2.5}$ above p50; daily $PM_{2.5}$ concentrations > 43

$\mu g/m^3$ for BTH, > 59 $\mu g/m^3$ for YRD, > 39 $\mu g/m^3$ for PRD) tend to occur when the circulation-based indices are negative. This is more often the case for heavily polluted days ($PM_{2.5}$ above p90; daily $PM_{2.5}$ concentrations > 97 $\mu g/m^3$ for BTH, > 110 $\mu g/m^3$ for YRD, > 68 $\mu g/m^3$ for PRD), in particular for BTH where 98% (44 of 45) of those days have negative values of $I_{Z500\_BTH}$ compared to 66% (119 of 180) of moderately polluted days (p50-90 $PM_{2.5}$). However, there is no such apparent distinction in the other two regions, since around 62% of both heavily and moderately polluted days in YRD have negative values of $I_{SLP\_YRD}$, and 70% of these days in PRD have negative values of $I_{V850\_PRD}$). Alternatively, 51% (23 of 45), 16% (7 of

45), 13% (6 of 45) of heavily polluted days in BTH, YRD and PRD, respectively, occur when circulation-based indices are below -1.

By contrast, moderately clean days ($PM_{2.5}$ below p50) and, to a greater extent, clean days ($PM_{2.5}$ below p10; daily $PM_{2.5}$ concentrations < 16 $\mu g/m^3$ for BTH, < 29 $\mu g/m^3$ for YRD, <15 $\mu g/m^3$ for PRD) tend to occur when circulation-based indices

are positive. 91% (41 of 45), 80% (36 of 45), 89% (40 of 45) of clean days in BTH, YRD and PRD have positive indices values. As expected, this tendency is even more pronounced for larger values of the indices, as 93% (52 of 56), 88% (37 of 42) and 93% (37 of 40) of days with $I_{Z500\_BTH}$, $I_{SLP\_YRD}$ and $I_{V850\_PRD}$ exceeding 1 are classified as moderately clean or clean. The share of days with positive values of the circulation indices generally decreases with increasing $PM_{2.5}$ pollution levels for all three regions, especially for PRD where the percentage of days with positive values of $I_{V850\_PRD}$ decreases from 89% of

clean days to only 61% (110 of 180) of moderately clean days. The results of the analyses conducted so far show that the daily circulation-based indices proposed in this study can capture most of the day-to-day variability of $PM_{2.5}$ and also identify days with different pollution levels, although with poorer performance for YRD than for the other two regions.

## 6 Joint effect of large-scale circulation and regional meteorology

The relatively moderate correlation between daily circulation-based index and daily $PM_{2.5}$ concentrations in YRD reflects the

complex mix of factors affecting the day-to-day variability of this pollutant. We have also found that regional meteorological variables (the most relevant ones identified in Section 3) influence the $PM_{2.5}$ concentrations for the three regions (e.g., $r$ = 0.66 for RH in BTH, $r$ = -0.43 for WSPD in YRD and $r$ = -0.52 for RH in PRD). On the other hand, there are significant correlations between the circulation-based indices and the most relevant regional meteorological variables in each region,



indicating that the effect of circulation on PM$_{2.5}$ occurs through modulation of the regional meteorology. The relationship

between the daily circulation-based index and the most important daily regional meteorological variable is weaker in YRD ($r$ = 0.29) than for the other two regions ($r$ = -0.64 for BTH; $r$ = 0.64 for PRD) (Table 2). This shows that the daily circulation-based index is not solely capable of capturing the regional meteorological variability driving day-to-day PM$_{2.5}$ changes in YRD.

While there is some co-variation of the large-scale circulation with the regional meteorology, they can be combined to

reproduce the day-to-day variability of PM$_{2.5}$ with improved performance. We therefore build multiple regression models including a linear combination of the most important regional meteorological field and the large-scale circulation index in each region (Table 3). The inclusion of regional meteorology explains more of the variance in the PM$_{2.5}$ concentrations for all three regions ($R^2$ ($I_{Z500\_BTH}$ + RH) = 0.54, $R^2$ ($I_{SLP\_YRD}$ + WSPD) = 0.23, $R^2$ ($I_{V850\_PRD}$ + RH) = 0.30) than the large-scale circulation index alone ($R^2$ ($I_{Z500\_BTH}$) = 0.45, $R^2$ ($I_{SLP\_YRD}$) = 0.11, $R^2$ ($I_{V850\_PRD}$) = 0.18). However, if we consider the regional

meteorological variable alone we see that its relationship with daily PM$_{2.5}$ concentrations explains more of the variance than the large-scale circulation variable for the YRD and PRD regions. Hence, compared to a linear model on the most relevant regional meteorological field, these multiple models do not bring major improvements for YRD and PRD, where the increase in explained variance is relatively small (0.18 vs. 0.23 for YRD and 0.27 vs. 0.30 for PRD). As expected, the signs of the regression coefficients for the most important regional meteorological field and the large-scale circulation index

(Table S2) are consistent with those of their respective correlation coefficients with PM$_{2.5}$.

**7 Discussion and conclusions**

This study investigates the modulation of daily PM$_{2.5}$ concentrations by regional meteorological conditions and large-scale circulation in three major populated regions of China during winter. Using a new high-resolution Chinese air quality reanalysis dataset, major regions associated with BTH, YRD and PRD are identified where daily PM$_{2.5}$ concentrations are

spatially coherent. For these three regions, we find that the regional meteorological variables most correlated with daily PM$_{2.5}$ concentrations are different: RH on the same day for BTH ($r$ = 0.66), WSPD one day before for YRD ($r$ = -0.43), and RH two days before for PRD ($r$ = -0.52). We identify the dominant large-scale circulation patterns associated with heavily polluted days (PM$_{2.5}$ above p90) considering the same time lags. Based on these dominant large-scale circulation features, we propose three new circulation-based indices that can be used both to explain the day-to-day variability of the PM$_{2.5}$

concentrations and to predict the occurrence of heavily polluted days and clean days (PM$_{2.5}$ below p10) in each region. The proposed circulation indices capture the relationship between the dominant large-scale circulation and daily PM$_{2.5}$ concentrations better than existing EAWM indices (Yang et al., 2002; Sun and Li, 1997; Jhun and Lee, 2004) and the Siberian High index (Wu and Wang, 2002). They improve on the capability of circulation-based indices (e.g., Wu and Wang, 2002; Huang et al., 2016) to distinguish PM$_{2.5}$ pollution levels in BTH, and provide the first daily circulation-based indices

specifically for YRD and PRD. Furthermore, the inclusion of regional meteorology slightly improves the performance of the




large-scale circulation-based indices to predict the evolution of the regional PM$_{2.5}$ concentrations in these regions on daily time scales.

Specifically, in BTH, heavily polluted days preferentially occur on days with a weak and eastward-extended Siberian High, weak northerly winds at 850 hPa over North China, and a shallow East Asian trough at 500 hPa. Of these meteorological

variables, we find that the shallow East Asian trough, represented as the negative area-weighted daily averages of Z500 centred over Northeast China, has the strongest relationship with both PM$_{2.5}$ concentrations ($r$ = -0.67) and RH ($r$ = -0.64). This suggests a strong contribution of the warm, humid air from the south and weak transport of northerly cold, dry air associated with the shallow East Asian trough to air pollution accumulation in BTH. Hence the intensity of the trough can be used as an indicator of PM$_{2.5}$ levels. The 500 hPa geopotential height-based index for BTH ($I_{Z500\_BTH}$) can distinguish

between different levels of PM$_{2.5}$ pollution well, not just between heavily polluted and clean conditions. On average, PM$_{2.5}$ pollution levels generally increase as the daily values of $I_{Z500\_BTH}$ turn from positive to negative. Furthermore, the inclusion of RH ($R^2$ $(I_{Z500\_BTH}$ + RH) = 0.54) explains more daily PM$_{2.5}$ variability than the simpler model $I_{Z500\_BTH}$ ($R^2$ $(I_{Z500\_BTH})$ = 0.45).

In YRD, heavily polluted days are characterised by the following circulation anomalies: a weak Siberian High, strong

northerly winds over North China and weak southerly winds over South China, as well as a shallow East Asian trough with positive Z500 anomalies centred over Japan. A weak Siberian High, represented as the area-weighted daily averages of SLP centred over Northeast China, shows the largest correlation with PM$_{2.5}$ concentrations ($r$ = -0.33) and WSPD ($r$ = 0.29), and hence is the dominant circulation variable used to build an index of PM$_{2.5}$ levels. This reflects the relationship between weak southerly winds over southern China, associated with a weak Siberian High, and poor horizontal dispersion of polluted air in

YRD. The sea level pressure-based index for YRD ($I_{SLP\_YRD}$) does not perform as well as $I_{Z500\_BTH}$ in distinguishing between moderately clean (p10-50 PM$_{2.5}$) and clean days or between moderately polluted (p50-90 PM$_{2.5}$) and heavily polluted days. The inclusion of WSPD ($R^2$ $(I_{SLP\_YRD}$ + WSPD) = 0.23) explains more daily PM$_{2.5}$ variability than $I_{SLP\_YRD}$ ($R^2$ $(I_{SLP\_YRD})$ = 0.11).

In PRD, heavily polluted days are characterised by La Niña-related circulation patterns: reduced SLP centred over northern

China and increased SLP over southwestern China and south-eastern Asia, weak southerly winds at 850 hPa over southern China, as well as a precipitation deficit over southern China. Weak southerly winds over southern China are identified as the dominant large-scale circulation anomaly, as seen from the largest correlations of the area-weighted daily averages of V850 over southern China and the South China Sea with PM$_{2.5}$ concentrations ($r$ = -0.43) and RH ($r$ = 0.64) in the region. This illustrates the influence of flow from more polluted continental regions and of precipitation deficits under weak humid

southerly winds on PM$_{2.5}$ pollution through reduced wet deposition in PRD. The 850 hPa meridional wind-based index for PRD ($I_{V850\_PRD}$) can distinguish between moderately clean and clean days better than between moderately polluted and heavily polluted days. The inclusion of RH ($R^2$ $(I_{V850\_PRD}$ + RH) = 0.30) explains considerably more daily PM$_{2.5}$ variability than $I_{V850\_PRD}$ ($R^2$ $(I_{V850\_PRD})$ = 0.18).



These results demonstrate the benefits of considering the large-scale circulation for air quality studies over China. The novel circulation indices proposed here explain a considerable fraction of the day-to-day variability of winter PM$_{2.5}$ over major metropolitan regions and can be combined with regional meteorological fields to improve our capability to predict the variability of this pollutant. Although the circulation indices explain less variance than the most relevant regional meteorological fields for YRD and PRD, we expect weather prediction and climate models to better represent these features of the large-scale circulation than regional meteorological fields such as surface wind speed and RH. There are however two limitations inherent in this work. Firstly, the relationships between atmospheric circulation and daily PM$_{2.5}$ concentrations may not be linear, as assumed in this study. Although we have improved the explained daily variability of PM$_{2.5}$ by linearly combining the most important regional meteorological field and the large-scale circulation index, non-linear models that account for the covariance of meteorological fields (e.g., Barmpadimos et al., 2011, 2012, Garrido-Perez et al., 2021) or dimensionality reduction techniques such as principal component analysis (e.g., Tai et al., 2012; Shen et al., 2015; Leung et al., 2018) merit further consideration. In addition, these large-scale relationships are based on only five winters of data, because high spatiotemporal coverage of air pollution measurements are only available in China from 2013. Hence, whilst our results are encouraging (e.g., we find that more than 80% of heavily polluted days in PRD occur in La Niña years), the robustness of these results need to be verified using longer-term data. Despite these limitations, this study has shown that dominant large-scale circulation features influence both regional meteorological processes and the PM$_{2.5}$ concentrations in BTH, YRD and PRD on a timescale of days, and can be used to construct indices for distinguishing clean days from heavily polluted days in these major populated regions of China. These results are beneficial to understanding and forecasting the occurrence of air pollution episodes in the three regions from a large-scale perspective.

**Acknowledgements**

We acknowledge the use of data from ERA-5 (https://cds.climate.copernicus.eu/cdsapp#!/dataset/reanalysis-era5-pressure-levels?tab=overview), GPCP (https://rda.ucar.edu/datasets/ds728.3/#!description) and CAQRA (http://www.en.scidb.cn/en/detail?dataSetId=696756084735475712&dataSetType=personal&version=V1#). Oliver Wild and Ruth M. Doherty thank the Natural Environment Research Council (NERC) for funding under grants NE/N006925/1, NE/N006976/1 and NE/N006941/1. Carlos Ordóñez thanks the Spanish Ministerio de Economía y Competitividad [grant number RYC-2014-15036]. Chaofan Li thanks the National Key Research and Development Program of China (Grant No. 2018YFA0606501).





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



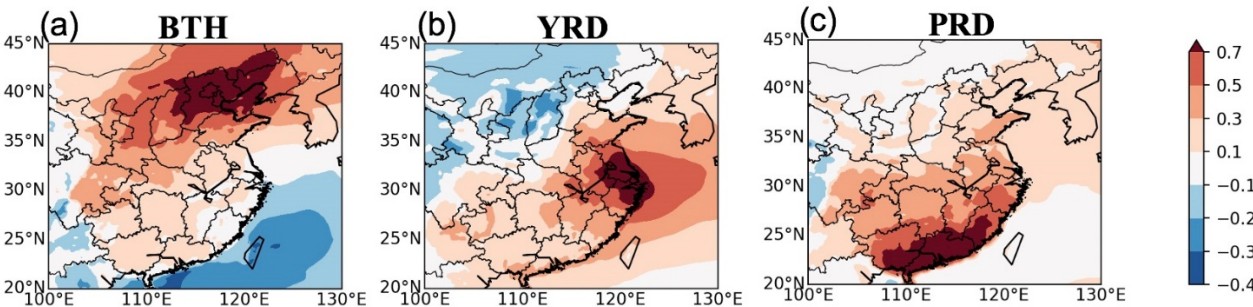

**Figure 1: Spatial correlation coefficients of daily mean PM$_{2.5}$ concentrations with those in (a) Beijing, (b) Shanghai and (c) Guangzhou during DJF 2013–2017. Regions where correlations are higher than 0.7 are selected to represent the Beijing–Tianjin–Hebei (BTH), Yangtze River Delta (YRD) and Pearl River Delta (PRD) regions, separately.**








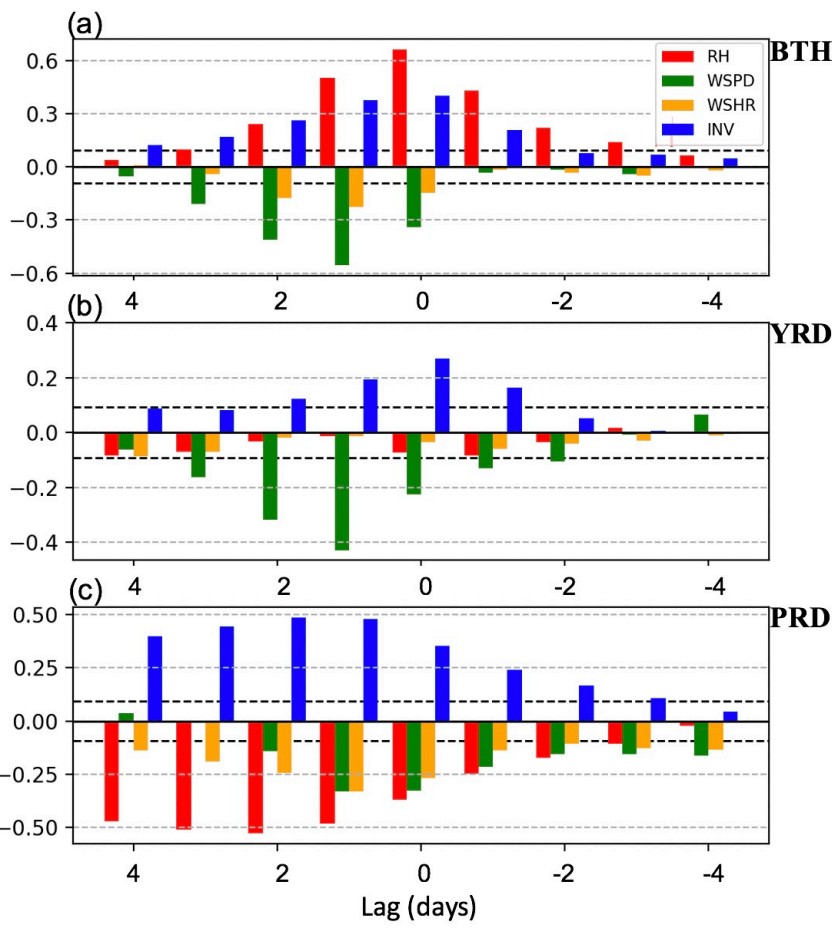

**Figure 2: Lagged correlations between daily mean PM$_{2.5}$ concentrations and regional meteorological variables including relative humidity (RH; red bars), wind speed (WSPD; green bars), vertical wind shear (WSHR; yellow bars) and inversion intensity (INV; blue bars) over (a) BTH, (b) YRD and (c) PRD during DJF 2013–2017. Black horizontal dashed lines indicate the 95% confidence level using the two-tailed Student's *t*-test.**







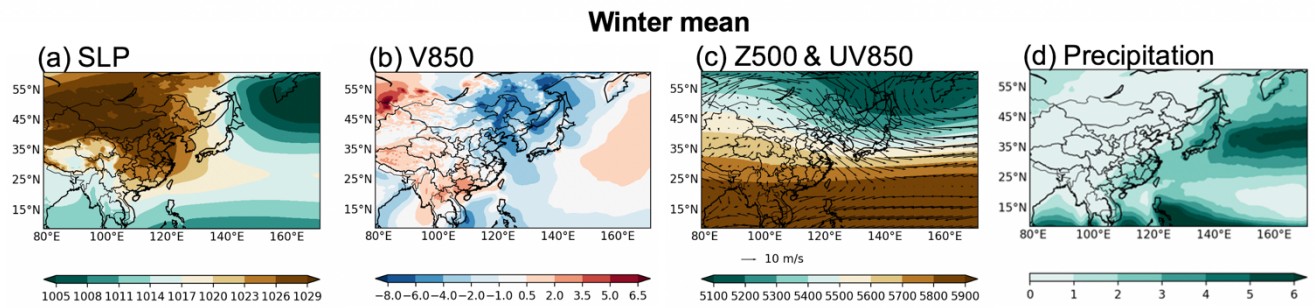

**Figure 3: Winter mean (a) sea level pressure (SLP; hPa), (b) 850-hPa meridional wind (V850; m s⁻¹), (c) 500-hPa geopotential height (Z500; m, shading) and 850 hPa wind (arrows), and (d) precipitation (mm day⁻¹) during DJF 2013–2017.**

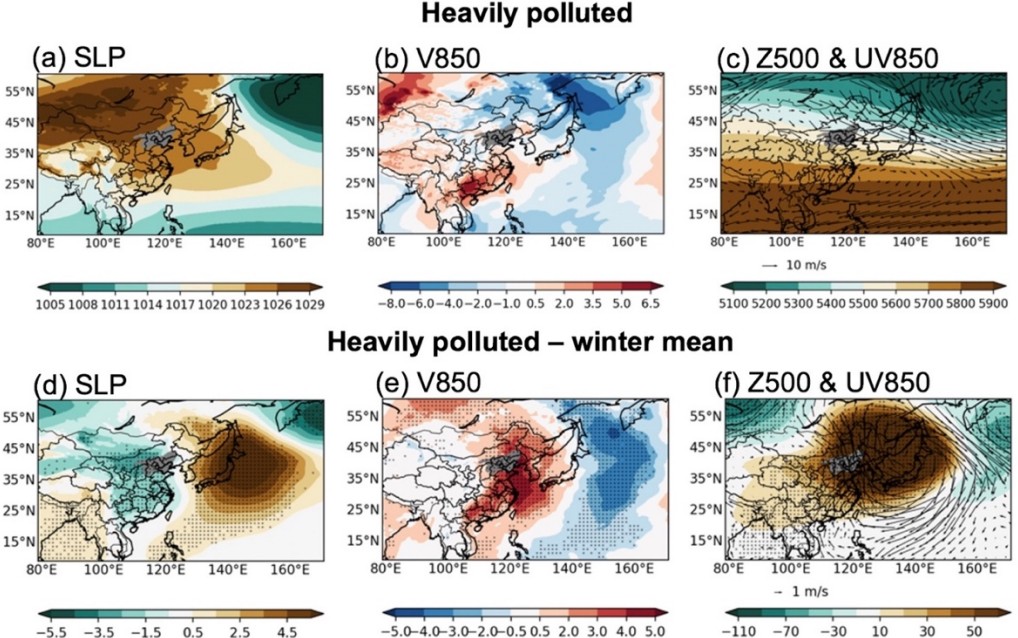

**Figure 4: Average (a) SLP (hPa), (b) V850 (m s⁻¹), (c) Z500 (m, shading) and 850 hPa wind (m s⁻¹, vector) on heavily polluted days (24-h PM$_{2.5}$ above the regional 90$^{th}$ percentile), and difference (heavily polluted days minus winter mean) for (d) SLP, (e) V850, (f) Z500 and 850 hPa wind during DJF 2013–2017 over BTH. For V850 (b, e), blue regions represent northerlies and red regions represent southerlies. Dotted regions in panels d–f mark statistically significant differences at the 95% level (determined through a bootstrap resampling method). Grey shading represents the BTH region.**



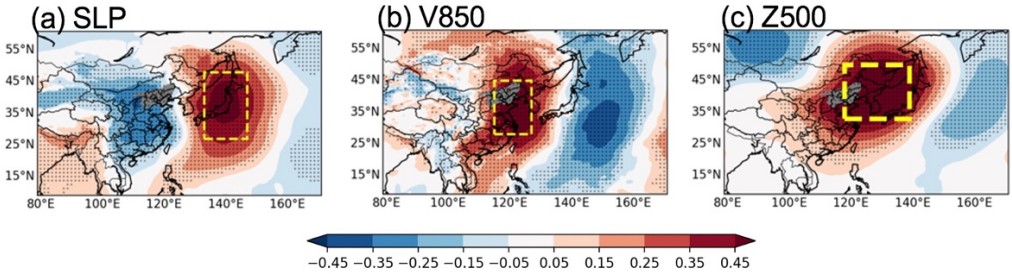

**Figure 5: Correlation coefficients of daily PM$_{2.5}$ concentrations in BTH with (a) SLP, (b) V850 and (c) Z500 during DJF 2013–2017 (dotted regions indicate significant correlations at the 95% level from the two-tailed Student's *t*-test). Grey shading represents the BTH region. The broad region presenting the highest correlation with BTH is marked by a yellow rectangle in each panel. The region used for the definition of a circulation-based index (eq. 3) is marked by a yellow thick rectangle in panel c.**

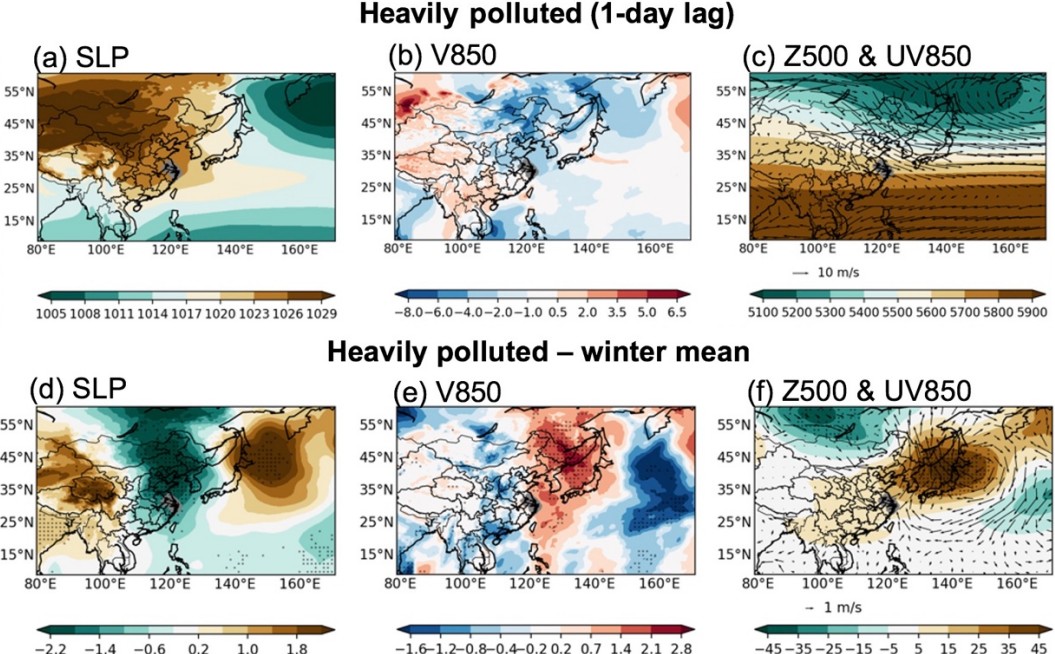

**Figure 6:** Average (a) SLP (hPa), (b) V850 (m s$^{-1}$), (c) Z500 (m, shading) and 850 hPa wind (m s$^{-1}$, vector) one day before heavily polluted days (24-h PM$_{2.5}$ above the regional 90$^{th}$ percentile), and difference (one day before heavily polluted days minus winter mean) for (d) SLP, (e) V850, (f) Z500 and 850 hPa wind during DJF 2013–2017 over YRD. For V850 (b, e), blue regions represent northerlies and red regions represent southerlies. Dotted regions in panels d–f mark statistically significant differences at the 95% level (determined through a bootstrap resampling method). Grey shading represents the YRD region.





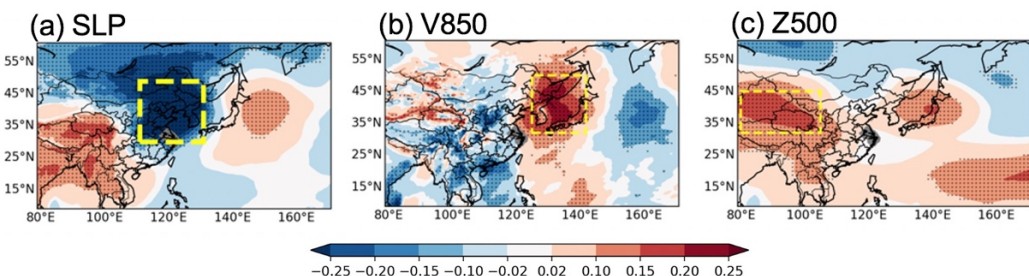

**Figure 7:** **Correlation coefficients of daily PM$_{2.5}$ concentrations in YRD with one day before (a) SLP, (b) V850 and (c) Z500 during DJF 2013–2017 (dotted regions indicate significant correlations at the 95% level from the two-tailed Student's *t*-test). Grey shading represents the YRD region. The broad region presenting the highest correlation with YRD is marked by a yellow rectangle in each panel. The region used for the definition of a circulation-based index (eq. 4) is marked by a yellow thick rectangle in panel a.**

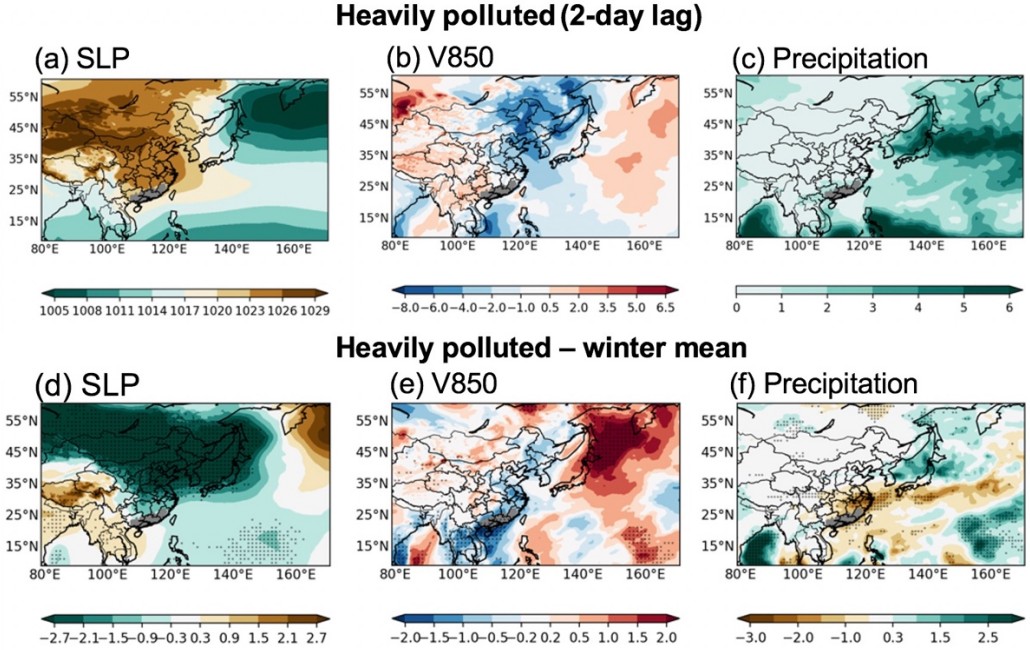

**Figure 8:** Average (a) SLP (hPa), (b) V850 (m s⁻¹) and (c) precipitation (mm day⁻¹) two days before heavily polluted days, and difference (two days before heavily polluted days minus winter mean) for (d) SLP, (e) V850, (f) precipitation during DJF 2013–2017 over PRD. For V850 (b, e), blue regions represent northerlies and red regions represent southerlies. Dotted regions in panels 4d–f mark statistically significant differences at the 95% level (determined through a bootstrap resampling method). Grey shading represents the PRD region.




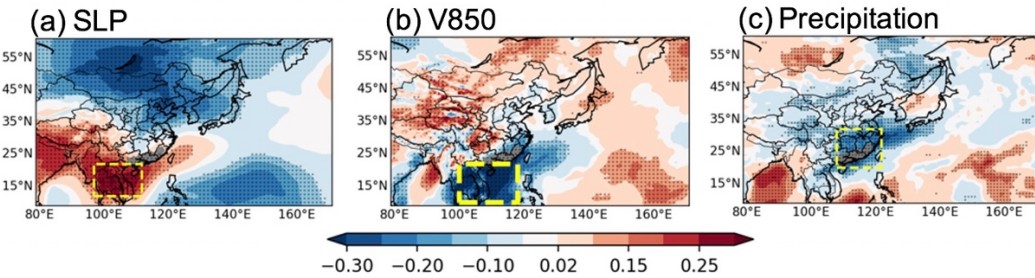

**Figure 9:** Correlation coefficients of daily PM$_{2.5}$ concentrations in PRD with two days before (a) SLP, (b) V850 and (c) precipitation during DJF 2013–2017 (dotted regions indicate significant correlations at the 95% level from the two-tailed Student's *t*-test). Grey shading represents the PRD region. The broad region presenting the highest correlation with PRD is marked by a yellow rectangle in each panel. The region used for the definition of a circulation-based index (eq. 5) is marked by a yellow thick rectangle in panel b.


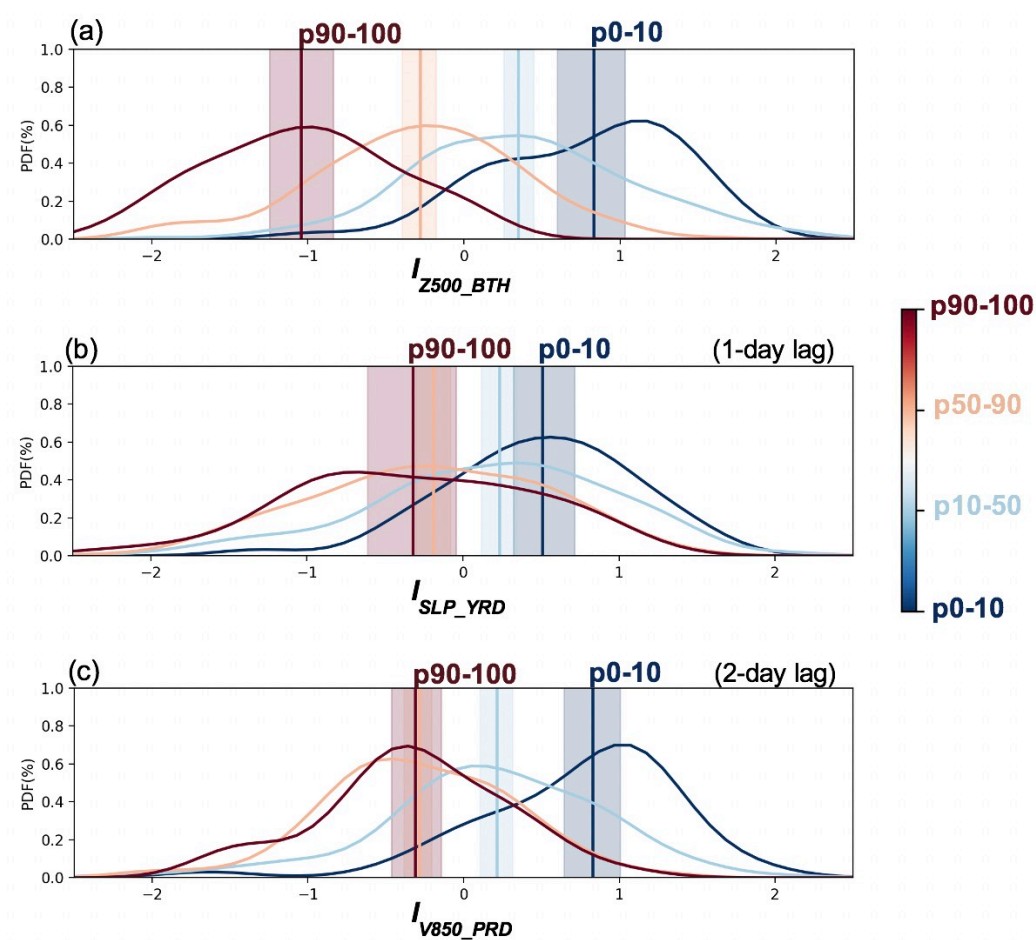

Figure 10: Frequency distributions of circulation-based indices for different percentile thresholds of daily mean PM$_{2.5}$ concentrations over (a) BTH, (b) YRD and (c) PRD during DJF 2013–2017. PM$_{2.5}$ concentration data are lagged by one and two days with respect to the circulation indices for YRD and PRD, respectively. The vertical lines and shading represent the averages and the associated 95% confidence intervals, respectively. Averages are calculated using Tukey's trimean (e.g., Ge et al., 2019): $\overline{X} = \frac{1}{4}(Q1 + 2Q2 + Q3)$, where $Q1$ is the lower quartile, $Q2$ is the median, and $Q3$ is the upper quartile. The confidence intervals for these averages are estimated by using bootstrap resampling (e.g., Wang, 2001). This method generates samples by randomly choosing daily values of circulation-based indices (resampling with replacement) and then calculating the Tukey's trimean. This process is repeated 10,000 times to get robust replicates of the mean. Ultimately, the lower and upper limits of the 95% confidence intervals are calculated as the values corresponding to the 2.5$^{th}$ and 97.5$^{th}$ percentiles.





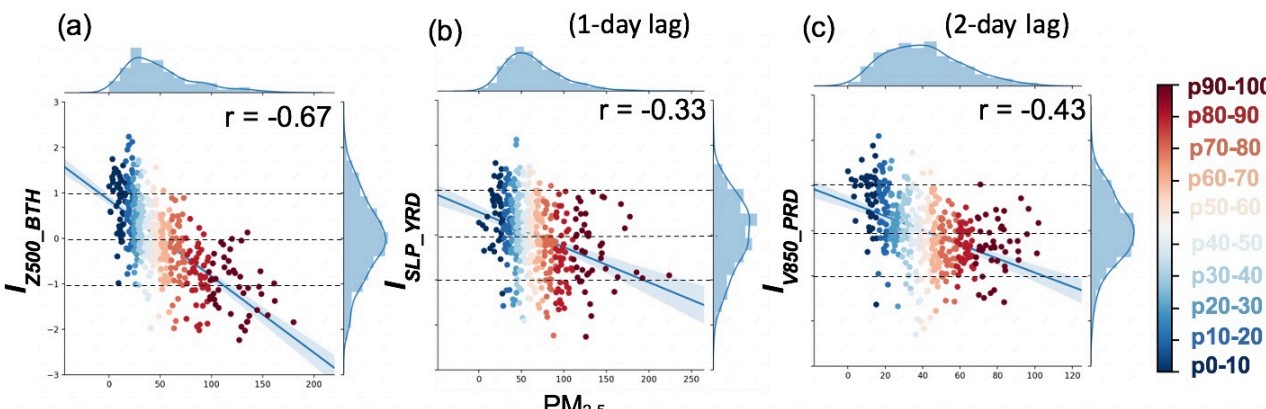

**Figure 11:** Joint distributions of circulation-based indices against detrended daily PM$_{2.5}$ concentrations for different percentile thresholds (colour coded), including the corresponding linear fits with 95% prediction intervals, over (a) BTH, (b) YRD and (c) PRD during DJF 2013–2017. PM$_{2.5}$ concentrations data are lagged by one and two days behind the circulation indices in the case of YRD and PRD, respectively.





**Table 1: Correlation of the area-weighted averages of daily normalized circulation variables over the regions marked by yellow rectangles in Figures 5, 7 and 9 with daily PM$_{2.5}$ concentrations over BTH, YRD and PRD, respectively, during DJF 2013–2017. All correlation values are significant at the 99% confidence level. The highest correlation (absolute value) for each region is shown in bold.**

| Correlation coefficient (450 days) | V850 | Z500 | SLP | Precipitation |
|---|---|---|---|---|
| PM$_{2.5}$ (BTH) | 0.59 | **0.67** | 0.54 | |
| PM$_{2.5}$ (YRD) | 0.25 | 0.21 | **-0.33** | |
| PM$_{2.5}$ (PRD) | **-0.43** | | 0.36 | -0.29 |





**Table 2: Correlation of circulation-based indices defined in this study (equations 3–5) with daily PM$_{2.5}$ concentrations over BTH, YRD, PRD, and with the most important regional meteorological variable in each region during DJF 2013–2017. All correlations are significant at the 99% confidence level.**

| Correlation coefficient (450 days) | $I_{Z500\_BTH}$ | $I_{SLP\_YRD}$ | $I_{V850\_PRD}$ |
|---|---|---|---|
| PM$_{2.5}$ | -0.67 | -0.33 | -0.43 |
| RH | -0.64 |  | 0.64 |
| WSPD |  | 0.29 |  |













**Table 3: Linear relationship (explained variance) of daily PM$_{2.5}$ concentrations in BTH, YRD and PRD, with the circulation-based index of Table 2, the most important regional meteorological field in each region and the linear combination of both during DJF 2013–2017. All the linear relationships are significant at the 99% confidence level.**

| BTH | $I_{Z500\_BTH}$ | RH | $I_{Z500\_BTH}$ + RH |
|---|---|---|---|
| $R^2$ | 0.45 | 0.44 | 0.54 |
| YRD | $I_{SLP\_YRD}$ | WSPD | $I_{SLP\_YRD}$ + WSPD |
| $R^2$ | 0.11 | 0.18 | 0.23 |
| PRD | $I_{V850\_PRD}$ | RH | $I_{V850\_PRD}$ + RH |
| $R^2$ | 0.18 | 0.27 | 0.30 |