# Peer review of "The impact of large-scale circulation on daily fine particulate matter $(PM_{2.5})$ over major populated regions of China in winter"

_Atmospheric Chemistry and Physics, 2021_

## Referee Comment (RC1)

**General Comments**

This manuscript aims to develop circulation-based indices to predict different levels of PM2.5 concentrations over three main regions in China - Beijing–Tianjin–Hebei (BTH), the Yangtze River Delta (YRD), and the Pearl River Delta (PRD). The manuscript is well written for the most part. The statistical analyses used to determine the influence of meteorology on PM2.5 are robust, with the assumptions and references clearly indicated. The proposed indices improve on the capability of circulation-based indices to distinguish PM2.5 pollution levels in BTH and provide the first daily circulation-based indices specifically for YRD and PRD.

I have some comments and questions, indicated below.

**Specific Comments**

(1) Page 2 Line 50: Temperature is also an important factor contributing to the variability in air quality and should be included in the paragraph.
(2) Page 3 Line 72: What differentiates the analyses presented in the current study from Leung et al., 2018 and Hou at al., 2019?
(3) Page 3 Line 91: Please follow the ERA5 guidelines to cite their datasets (https://confluence.ecmwf.int/pages/viewpage.action?pageId=197704114)
(4) Section 2: The datasets used in the study (ERA5, CAQRA, GPCP) are all at different spatial resolutions. How is this accounted for in the analysis?
(5) Page 5 Line 155-160 and Figure 2: Why is there a non-significant correlation between $PM_{2.5}$ and RH in YRD?
(6) Page 6 Line 165: Formatting ($PM_{2.5}$ does not appear in-line with the other text)
(7) Page 6 Line 169: Remind the reader what SLP stands for
(8) Page 6: Line 186: The author mentions –

"Considering the moderate correlations found for YRD and PRD, we further investigate the influence of large-scale circulation on daily PM2.5 variability through its direct effect on the most important regional meteorological variables identified separately for the three regions."

However, considering that the absolute correlation coefficients do not exceed 0.12 for YRD and PRD (line 182 Page 6), please re-word this sentence to reflect that the correlations are "low".

(9) Section 4.3: Why are the circulation variables used here (PRD) different from the other two regions?

(10) My main question is – if the most relevant meteorological fields explain more variance than the circulation indices, what is the value in using these indices? The authors briefly answer this in the last paragraph: **"Although the circulation indices explain less variance than the most relevant regional meteorological fields for YRD and PRD, we expect weather**

**prediction and climate models to better represent these features of the large-scale circulation than regional meteorological fields such as surface wind speed and RH."**

I would encourage the authors to elaborate on their argument and present more evidence for this – why would weather prediction and climate models better represent these indices over the meteorological fields?

(11)     Finally, I found the last section "Discussion and Conclusions" to be too long. I would recommend cutting it down to a paragraph or two, to provide a succinct summary of the main findings from their analysis.

---

## Referee Comment (RC2)

This study identified the dominant large-scale circulation associated with heavily polluted days for BTH, YRD and PRD through its effect on the most important regional meteorological variables, and propose specific circulation-based indices for these three regions. This work is meaningful. But there are still some places needing to improve.

1.The study reminded many times about the word "first", but it is really not the first one to do these investigations. Such Hou et al. (2018, 2020), and so on. Hou et al. (2018) also did many statistical analyses about the indictors. Please cites them.

Hou, X. W., D. D. Fei, H. Q. Kang, Y. L. Zhang, J. H. Gao, (2018). Seasonal statistical analysis of the impact of meteorological factors on fine particle pollution in China in 2013–2017, Nat. Hazards, https://doi.org/10.1007/s11069-018-3315-y.

Hou, X., Zhu, B., Kumar, K. R., de Leeuw, G., Lu, W., Huang, Q., & Zhu, X. (2020). Establishment of conceptual schemas of surface synoptic meteorological situations affecting fine particulate pollution across eastern China in the winter. Journal of Geophysical Research: Atmospheres, 125, e2020JD033153. https://doi.org/10.1029/2020JD033153.

2. The dataset used in the study is very important. It determines the credibility of your work. Please add the simply introduction about the dataset in the section of Abstract.

3. About Figure 1, please give more detail description about the classification of three regions.

4. Discussion and conclusions should be a summary of the study. Please make the sentence more concentrated.

---

## Author Comment (AC1)

Dear ACP editor and reviewers,

We thank both reviewers for their positive comments and constructive suggestions. Below we provide our point by point replies to the comments (**in bold**).

**Referee: 1**

**(1) Page 2 Line 50: Temperature is also an important factor contributing to the variability in air quality and should be included in the paragraph.**

We have included it in this paragraph as you suggest (revised manuscript page 2, lines 16-18, 22):

*"...b) sulphate and secondary organic aerosol formation and the volatilization of ammonium nitrate and semi-volatile organics favoured by high temperature (Dawson et al., 2007; Aksoyoglu et al., 2011)...Specifically, high temperature and RH, weak WSPD, strong INV and weak WSHR have been found to contribute to the accumulation and growth of pollutants in a shallow and stable boundary layer over the North China Plain..."*

**(2) Page 3 Line 72: What differentiates the analyses presented in the current study from Leung et al., 2018 and Hou at al., 2019?**

Leung et al. (2018) did a principal component analysis of different regional and large-to-synoptic scale fields to provide some distinct meteorological modes of $PM_{2.5}$ variability over each region. Our study identifies the most important regional meteorological variables first, and then diagnoses the dominant large-scale circulation associated with heavily polluted days through its effect on the regional meteorology. Based on these dominant large-scale circulation – $PM_{2.5}$ relationships, only one large-scale field is used to define an index for each region. Because of this, our indices are simpler and therefore probably easier to derive and apply than theirs. Furthermore, our study considers some regional meteorological variables related to the vertical ventilation (i.e., wind shear and inversion intensity) and the mid-level large-scale circulation (i.e., the middle tropospheric East Asian trough), which are not addressed in Leung et al. (2018).

Hou et al. (2019) identified which of four weather types classified from sea level pressure and 10-m wind is most likely to be responsible for the occurrence of high $PM_{2.5}$ concentration over BTH, YRD and PRD. Our study considers other large-scale and regional-scale meteorological variables. Besides that, more importantly, they did not further propose a large-scale circulation index for each region as our study does. We have briefly mentioned this in the main text (revised manuscript page 3, lines 17-19):

*"Indeed, Leung et al. (2018) found that different distinct meteorological modes could explain the variability of $PM_{2.5}$ in BTH, YRD and PRD, but simple large-scale circulation indices have not been defined for the latter two regions as yet."*

**(3) Page 3 Line 91: Please follow the ERA5 guidelines to cite their datasets** (https://confluence.ecmwf.int/pages/viewpage.action?pageId=197704114)

We have cited ERA5 following the guidelines (revised manuscript page 4, lines 2-4):

*"We use daily meteorological data from the fifth-generation atmospheric reanalysis ERA5 provided by the European Centre for Medium-Range Weather Forecasts at a spatial resolution of 0.25° (Copernicus Climate Change Service [C3S], 2017; Hersbach et al., 2020)."*

**(4) Section 2: The datasets used in the study (ERA5, CAQRA, GPCP) are all at different spatial resolutions. How is this accounted for in the analysis?**

This is a good point but we think it is not a major issue. As indicated by the referee, we use different datasets: CAQRA at ~0.13° horizontal resolution to represent the $PM_{2.5}$ concentrations and identify the three regions of study, GPCP at 1° resolution for precipitation and ERA5 at 0.25° resolution for the rest of the meteorological fields. Although these meteorological fields are provided at coarser resolutions than that of the $PM_{2.5}$ concentrations, we do not feel this is problematic because they are mostly used to represent the large-scale circulation.

We also average some ERA5 fields over the three regions of analysis to investigate the relationship of the $PM_{2.5}$ concentrations with the meteorology at the regional scale (see e.g., Fig. 2). For such analyses, we consider the ERA5 grid cells that fall within the boundaries of the regions. Again, this is not a major issue because both ERA5 and CAQRA datasets have similar resolutions, and we investigate the regional (rather than local) signatures of air pollution over wide regions (those with dark red shading in Fig. 1).

**(5) Page 5 Line 155-160 and Figure 2: Why is there a non-significant correlation between PM2.5 and RH in YRD?**

In that part of the text, we indicate that the response of $PM_{2.5}$ to relative humidity (RH) differs by region. In northern China, especially over BTH, high RH contributes to high $PM_{2.5}$ pollution levels (significant positive correlation; Fig. 2a) due to the general contrast between clean, dry air reaching BTH from the northwest and more polluted, humid air reaching BTH from central and eastern China. In southern China, especially over PRD, high RH with clean oceanic air and precipitation facilitates the removal of aerosols by wet deposition (significant negative correlation; Fig. 2c). YRD can possibly therefore be considered as a transition region that is in the middle of northern and southern China, where the response of $PM_{2.5}$ to RH is affected by all the above processes (Fig. 2b; Leung et al., 2018; He et al., 2019). This may explain the absence of a significant correlation.

**(6) Page 6 Line 165: Formatting (PM2.5 does not appear in-line with the other text)**

We have modified the format (revised manuscript page 6, line 15).

**(7) Page 6 Line 169: Remind the reader what SLP stands for**
We have reminded the reader what SLP stands for, and V850, Z500 (revised manuscript page 6, lines 19-23):

*"Using ERA-5 reanalysis data for DJF 2013-17, we find the wintertime large-scale circulation over East Asia is dominated by the Siberian High as seen from the high sea level pressure (SLP) values centred over northwestern Mongolia (Fig. 3a). The Siberian High*

*induces northerly near-surface winds along its eastern edge, which bring cold, clean air to northern and central China as indicated by negative values of meridional wind at 850 hPa (V850) (Fig. 3b). This northerly near-surface flow is also associated with the middle tropospheric East Asian trough, characterised by low geopotential heights at 500 hPa (Z500) over Northeast China as seen in Figure 3c."*

**(8) Page 6: Line 186: The author mentions –**

**"Considering the moderate correlations found for YRD and PRD, we further investigate the influence of large-scale circulation on daily PM2.5 variability through its direct effect on the most important regional meteorological variables identified separately for the three regions."**

**However, considering that the absolute correlation coefficients do not exceed 0.12 for YRD and PRD (line 182 Page 6), please re-word this sentence to reflect that the correlations are "low".**

This sentence is now more explicit (revised manuscript page 7, lines 6):

*"As the correlations of the daily PM$_{2.5}$ concentrations with the mentioned indices are low for YRD and PRD, we further…"*

**(9) Section 4.3: Why are the circulation variables used here (PRD) different from the other two regions?**

We examine the relationship of PM$_{2.5}$ with both SLP and 850 hPa wind in all regions. The only difference is that Z500 is examined for both BTH (Figs. 4–5) and YRD (Figs. 6–7) but not for PRD (Figs. 8–9) where precipitation is examined instead. Before examining the role of atmospheric circulation, we identify that high relative humidity (RH), weak surface wind speed and low RH contribute most to high PM$_{2.5}$ levels in BTH, YRD and PRD, respectively in Section 3. Both high RH over BTH and weak surface wind speed over YRD are associated with inhibited northerly cold air over northern and central China which has been found to be favoured by a shallow East Asian trough (characterised by low Z500 values) (e.g., Zhang et al., 2014). Unlike BTH, RH is negatively correlated with PM$_{2.5}$ concentrations over PRD and the high correlations persist for several days (Fig. 2). The strong association of high RH with precipitation over southern China has been found to facilitate PM$_{2.5}$ wet deposition in several studies (e.g., Zhu et al., 2012; Leung et al., 2018). Therefore, we examine Z500 for both BTH and YRD, and precipitation for PRD in Section 4.3.

**(10) My main question is – if the most relevant meteorological fields explain more variance than the circulation indices, what is the value in using these indices? The authors briefly answer this in the last paragraph: "Although the circulation indices explain less variance than the most relevant regional meteorological fields for YRD and PRD, we expect weather prediction and climate models to better represent these features of the large-scale circulation than regional meteorological fields such as surface wind speed and RH."**

**I would encourage the authors to elaborate on their argument and present more evidence for this – why would weather prediction and climate models better represent these indices over the meteorological fields?**

Thank you for pointing this out. In this study, surface wind speed and relative humidity (RH) are identified as the most important regional meteorological fields for YRD and PRD, respectively. However, current climate models have a relatively weak capability to represent some regional signals (e.g., Chen et al. 2012). For instance, almost all CMIP5 models exhibit lower interannual variability of surface wind speed over eastern China than reanalysis data and observations (Zha et al., 2020). Furthermore, RH is significantly overestimated in most CMIP6 models (Xu et al., 2021). Such discrepancies may at least partly arise from the underrepresentation of subgrid scale processes in climate models. Therefore, we expect climate models to better represent the large-scale circulation than regional meteorological fields.

On the other hand, climate model output can be exploited to project the inter-annual variability, decadal oscillation and long-term trends of relevant circulation indices under climate change in a relatively simple manner. These can be used to understand the future evolution of $PM_{2.5}$ (e.g., Cai et al., 2017; Zhao et al., 2021) and guide contemporary strategies for emission reduction. We have now mentioned this in the main text (revised manuscript page 14, lines 2-10):

*"Although the circulation indices explain less variance than the most relevant regional meteorological fields for YRD and PRD, we expect climate models to represent these features of the large-scale circulation better than regional meteorological fields that depend on subgrid scale processes. Indeed, current climate models have a limited capability to represent some regional signals (e.g., RH: Xu et al., 2021; surface wind speed: Zha et al., 2020). On the other hand, climate model projections of the inter-annual variability, decadal oscillations and long-term trends of circulation indices are appropriate to represent the future evolution of the $PM_{2.5}$ concentrations under climate change (e.g., Cai et al., 2017; Zhao et al., 2021), considering different degrees of pollution control. Such an approach could be applied to guide air quality policies aimed at keeping future $PM_{2.5}$ concentrations below current levels."*

**(11) Finally, I found the last section "Discussion and Conclusions" to be too long. I would recommend cutting it down to a paragraph or two, to provide a succinct summary of the main findings from their analysis.**

We have made the summary text at the beginning of this section more succinct and cut it down to two paragraphs. Note, however, that some discussion is still provided after the summary to highlight both the usefulness (see reply to previous comment) and the limitations of this study.